# Deep learning for universal linear embeddings of nonlinear dynamics

Bethany Lusch [1,2], J. Nathan Kutz[1] & Steven L. Brunton[1,2]

Identifying coordinate transformations that make strongly nonlinear dynamics approximately linear has the potential to enable nonlinear prediction, estimation, and control using linear theory. The Koopman operator is a leading data-driven embedding, and its eigenfunctions provide intrinsic coordinates that globally linearize the dynamics. However, identifying and representing these eigenfunctions has proven challenging. This work leverages deep learning to discover representations of Koopman eigenfunctions from data. Our network is parsimonious and interpretable by construction, embedding the dynamics on a low-dimensional manifold. We identify nonlinear coordinates on which the dynamics are globally linear using a modified auto-encoder. We also generalize Koopman representations to include a ubiquitous class of systems with continuous spectra. Our framework parametrizes the continuous frequency using an auxiliary network, enabling a compact and efficient embedding, while connecting our models to decades of asymptotics. Thus, we benefit from the power of deep learning, while retaining the physical interpretability of Koopman embeddings.

[1] Department of Applied Mathematics, University of Washington, Seattle, WA 98195, USA. [2] Department of Mechanical Engineering, University of Washington, Seattle, WA 98195, USA. Correspondence and requests for materials should be addressed to B.L. (email: herwaldt@uw.edu)

Nonlinearity is a hallmark feature of complex systems, giving rise to a rich diversity of observed dynamical behaviors across the physical, biological, and engineering sciences[1,2]. Although computationally tractable, there exists no general mathematical framework for solving nonlinear dynamical systems. Thus representing nonlinear dynamics in a linear framework is particularly appealing because of powerful and comprehensive techniques for the analysis and control of linear systems[3], which do not readily generalize to nonlinear systems. Koopman operator theory, developed in 1931[4,5], has recently emerged as a leading candidate for the systematic linear representation of nonlinear systems[6,7]. This renewed interest in Koopman analysis has been driven by a combination of theoretical advances[6–10], improved numerical methods such as dynamic mode decomposition (DMD)[11–13], and an increasing abundance of data. Eigenfunctions of the Koopman operator are now widely sought, as they provide intrinsic coordinates that globally linearize nonlinear dynamics. Despite the immense promise of Koopman embeddings, obtaining representations has proven difficult in all but the simplest systems, and representations are often intractably complex or are the output of uninterpretable black-box optimizations. In this work, we utilize the power of deep learning for flexible and general representations of the Koopman operator, while enforcing a network structure that promotes parsimony and interpretability of the resulting models.

Neural networks (NNs), which form the theoretical architecture of deep learning, were inspired by the primary visual cortex of cats where neurons are organized in hierarchical layers of cells to process visual stimulus[14]. The first mathematical model of a NN was the neocognitron[15] which has many of the features of modern deep neural networks (DNNs), including a multi-layer structure, convolution, max pooling, and nonlinear dynamical nodes. Importantly, the universal approximation theorem[16–18] guarantees that a NN with sufficiently many hidden units and a linear output layer is capable of representing any arbitrary function, including our desired Koopman eigenfunctions. Although NNs have a four-decade history, the analysis of the ImageNet data set[19], containing over 15 million labeled images in 22,000 categories, provided a watershed moment[20]. Indeed,

powered by the rise of big data and increased computational power, deep learning is resulting in transformative progress in many data-driven classification and identification tasks[19–21]. A strength of deep learning is that features of the data are built hierarchically, which enables the representation of complex functions. Thus, deep learning can accurately fit functions without hand-designed features or the user choosing a good basis[21]. However, a current challenge in deep learning research is the identification of parsimonious, interpretable, and transferable models[22].

Deep learning has the potential to enable a scaleable and data-driven architecture for the discovery and representation of Koopman eigenfunctions, providing intrinsic linear representations of strongly nonlinear systems. This approach alleviates two key challenges in modern dynamical systems: (1) equations are often unknown for systems of interest[23–25], as in climate, neuroscience, epidemiology, and finance; and, (2) low-dimensional dynamics are typically embedded in a high-dimensional state space, requiring scaleable architectures that discover dynamics on latent variables. Although NNs have also been used to model dynamical systems[26] and other physical processes[27] for decades, great strides have been made recently in using DNNs to learn Koopman embeddings, resulting in several excellent papers[28–33]. For example, the VAMPnet architecture[28,29] uses a time-lagged auto-encoder and a custom variational score to identify Koopman coordinates on an impressive protein folding example. In all of these recent studies, DNN representations have been shown to be more flexible and exhibit higher accuracy than other leading methods on challenging problems.

The focus of this work is on developing DNN representations of Koopman eigenfunctions that remain interpretable and parsimonious, even for high-dimensional and strongly nonlinear systems. Our approach (see Fig. 1) differs from previous studies, as we are focused specifically on obtaining parsimonious models that match the intrinsic low-rank dynamics while avoiding overfitting and remaining interpretable, thus merging the best of DNN architectures and Koopman theory. In particular, many dynamical systems exhibit a continuous eigenvalue spectrum, which confounds low-dimensional representation using existing

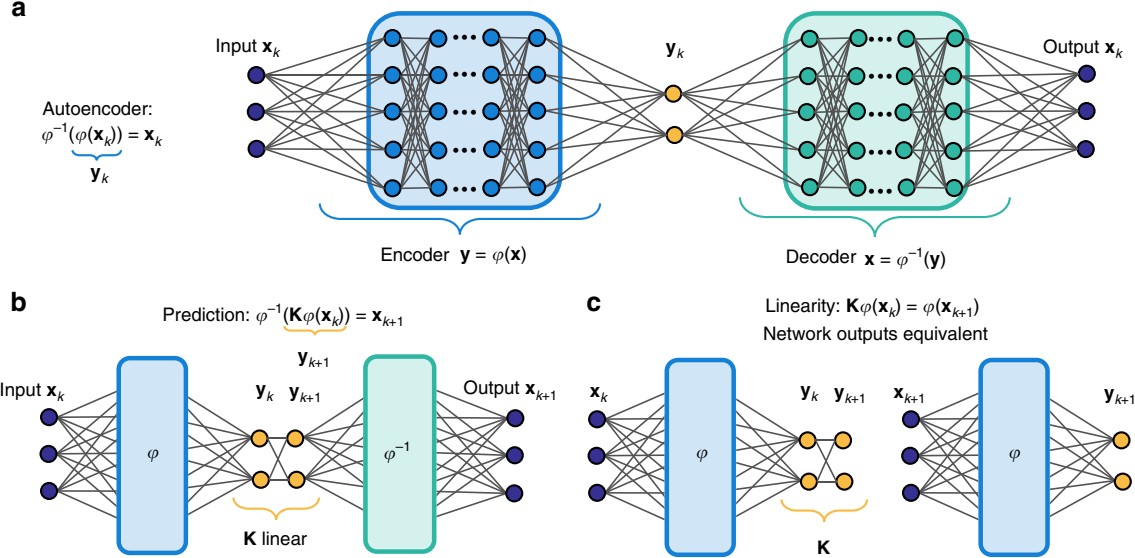

**Fig. 1** Diagram of our deep learning schema to identify Koopman eigenfunctions $\varphi(\mathbf{x})$. **a** Our network is based on a deep auto-encoder, which is able to identify intrinsic coordinates $\mathbf{y} = \varphi(\mathbf{x})$ and decode these coordinates to recover $\mathbf{x} = \varphi^{-1}(\mathbf{y})$. **b, c** We add an additional loss function to identify a linear Koopman model $\mathbf{K}$ that advances the intrinsic variables $\mathbf{y}$ forward in time. In practice, we enforce agreement with the trajectory data for several iterations through the dynamics, i.e. $\mathbf{K}^m$. In **b**, the loss function is evaluated on the state variable $\mathbf{x}$ and in **c** it is evaluated on $y$

DNN or Koopman representations. This work develops a generalized framework and enforces new constraints specifically designed to extract the fewest meaningful eigenfunctions in an interpretable manner. For systems with continuous spectra, we utilize an augmented network to parameterize the linear dynamics on the intrinsic coordinates, avoiding an infinite asymptotic expansion in harmonic eigenfunctions. Thus, the resulting networks remain parsimonious, and the few key eigenfunctions are interpretable. We demonstrate our deep learning approach to Koopman on several examples designed to illustrate the strength of the method, while remaining intuitive in terms of classic dynamical systems.

## Results

**Data-driven dynamical systems.** To give context to our deep learning approach to identify Koopman eigenfunctions, we first summarize highlights and challenges in the data-driven discovery of dynamics. Throughout this work, we will consider discrete-time dynamical systems

$$\mathbf{x}_{k+1} = \mathbf{F}(\mathbf{x}_k), \tag{1}$$

where $\mathbf{x} \in \mathbb{R}^n$ is the state of the system and $\mathbf{F}$ represents the dynamics that map the state of the system forward in time. Discrete-time dynamics often describe a continuous-time system that is sampled discretely in time, so that $\mathbf{x}_k = \mathbf{x}(k\Delta t)$ with sampling time $\Delta t$. The dynamics in $\mathbf{F}$ are generally nonlinear, and the state $\mathbf{x}$ may be high dimensional, although we typically assume that the dynamics evolve on a low-dimensional attractor governed by persistent coherent structures in the state space[2]. Note that $\mathbf{F}$ is often unknown and only measurements of the dynamics are available.

The dominant geometric perspective of dynamical systems, in the tradition of Poincaré, concerns the organization of trajectories of Eq. (1), including fixed points, periodic orbits, and attractors. Formulating the dynamics as a system of differential equations in $x$ often admits compact and efficient representations for many natural systems[25]; for example, Newton's second law is naturally expressed by Eq. (1). However, the solution to these dynamics may be arbitrarily complicated, and possibly even irrepresentable, except for special classes of systems. Linear dynamics, where the map $\mathbf{F}$ is a matrix that advances the state $x$, are among the few systems that admit a universal solution, in terms of the eigenvalues and eigenvectors of the matrix $\mathbf{F}$, also known as the spectral expansion.

**Koopman operator theory.** In 1931, B.O. Koopman provided an alternative description of dynamical systems in terms of the evolution of functions in the Hilbert space of possible measurements $\mathbf{y} = \mathbf{g}(\mathbf{x})$ of the state[4]. The so-called Koopman operator, $\mathcal{K}$, that advances measurement functions is an infinite-dimensional linear operator:

$$\mathcal{K}\mathbf{g} \triangleq \mathbf{g} \circ \mathbf{F} \quad \Rightarrow \quad \mathcal{K}\mathbf{g}(\mathbf{x}_k) = \mathbf{g}(\mathbf{x}_{k+1}). \tag{2}$$

Koopman analysis has gained significant attention recently with the pioneering work of Mezic et al.[6–10], and in response to the growing wealth of measurement data and the lack of known equations for many systems[13,25]. Representing nonlinear dynamics in a linear framework, via the Koopman operator, has the potential to enable advanced nonlinear prediction, estimation, and control using the comprehensive theory developed for linear systems. However, obtaining finite-dimensional approximations of the infinite-dimensional Koopman operator has proven challenging in practical applications.

Finite-dimensional representations of the Koopman operator are often approximated using the DMD[12,13], introduced by Schmid[11]. By construction, DMD identifies spatio-temporal coherent structures from a high-dimensional dynamical system, although it does not generally capture nonlinear transients since it is based on linear measurements of the system, $\mathbf{g}(\mathbf{x}) = \mathbf{x}$. Extended DMD (eDMD) and the related variational approach of conformation dynamics (VAC)[34–36] enriches the model with nonlinear measurements[33,37]. It has recently been shown that eDMD is equivalent to the variational approach of conformation dynamics (VAC)[34–36], first derived by Noé and Nüske in 2013 to simulate molecular dynamics with a broad separation of time-scales. Further connections between eDMD and VAC and between DMD and the time lagged independent component analysis (TICA) are explored in a recent review[38]. A key contribution of VAC is a variational score enabling the objective assessment of Koopman models via cross-validation. Recently, eDMD has been demonstrated to improve model predictive control performance in nonlinear systems[39].

Identifying regression models based on nonlinear measurements will generally result in closure issues, as there is no guarantee that these measurements form a Koopman invariant subspace[40]. The resulting models are of exceedingly high dimension, and when kernel methods are employed[41], the models may become uninterpretable. Instead, many approaches seek to identify eigenfunctions of the Koopman operator directly, satisfying:

$$\varphi(\mathbf{x}_{k+1}) = \mathcal{K}\varphi(\mathbf{x}_k) = \lambda\varphi(\mathbf{x}_k). \tag{3}$$

Eigenfunctions are guaranteed to span an invariant subspace, and the Koopman operator will yield a matrix when restricted to this subspace[40,42]. In practice, Koopman eigenfunctions may be more difficult to obtain than the solution of (1); however, this is a one-time up-front cost that yields a compact linear description. The challenge of identifying and representing Koopman eigenfunctions provides strong motivation for the use of powerful emerging deep learning methods[28–33].

**Koopman for systems with continuous spectra.** The Koopman operator provides a global linearization of the dynamics. The concept of linearizing dynamics is not new, and locally linear representations are commonly obtained by linearizing around fixed points and periodic orbits[1]. Indeed, asymptotic and perturbation methods have been widely used since the time of Newton to approximate solutions of nonlinear problems by starting from the exact solution of a related, typically linear problem. The classic pendulum, for instance, satisfies the differential equation $\ddot{x} = -\sin(\omega x)$ and has eluded an analytic solution since its mathematical inception. The linear problem associated with the pendulum involves the small angle approximation whereby $\sin(\omega x) = \omega x - (\omega x)^3/3! + \ldots$; and only the first term is retained in order to yield exact sinusoidal solutions. The next correction involving the cubic term gives the Duffing equation, which is one of the most commonly studied nonlinear oscillators in physics[1]. Importantly, the cubic contribution is known to shift the linear oscillation frequency of the pendulum, $\omega \rightarrow \omega + \Delta\omega$, as well as generate harmonics such as $\exp(\pm 3i\omega)$[43,44]. An exact representation of the solution can be derived in terms of Jacobi elliptic functions, which have a Taylor series representation in terms of an infinite sum of sinusoids with frequencies $(2n - 1)\omega$, where $n = 1,2,\ldots,\infty$. Thus, the simple pendulum oscillates at the (linear) natural frequency $\omega$ for small deflections, and as the pendulum energy is increased, the frequency decreases continuously, resulting in a so-called continuous spectrum.

The importance of accounting for the continuous spectrum was discussed in 1932 in an extension by Koopman and von Neumann[5]. A continuous spectrum, as described for the simple pendulum, is characterized by a continuous range of observed frequencies, as opposed to the discrete spectrum consisting of isolated, fixed frequencies. This phenomena is observed in a wide range of physical systems that exhibit broadband frequency content, such as turbulence and nonlinear optics. The continuous spectrum thus confounds simple Koopman descriptions, as there is not a straightforward finite approximation in terms of a small number of eigenfunctions[10]. Indeed, away from the linear regime, an infinite Fourier sum is required to approximate the shift in frequency and eigenfunctions. In fact, in some cases, eigenfunctions may not exist at all.

Recently, there have been several algorithmic advances to approximate systems with continuous spectra, including non-linear Laplacian spectral analysis[45] and the use of delay coordinates[46,47]. A critically enabling innovation of the present work is explicitly accounting for the parametric dependence of the Koopman operator $K(\lambda)$ on the continuously varying $\lambda$, related to the classic perturbation results above. By constructing an auxiliary network (see Fig. 2) to first determine the parametric dependency of the Koopman operator on the frequency $\lambda_\pm = \pm i\omega$, an interpretable low-rank model of the intrinsic dynamics can then be constructed. In particular, a nonlinear oscillator with continuous spectrum may now be represented as a single pair of conjugate eigenfunctions, mapping trajectories into perfect sines and cosines, with a continuous eigenvalue parameterizing the frequency. If this explicit frequency dependence is unaccounted for, then a high-dimensional network is necessary to account for the shifting frequency and eigenvalues. We conjecture that previous Koopman models using high-dimensional DNNs represent the harmonic series expansion required to approximate the continuous spectrum for systems such as the Duffing oscillator.

**Deep learning to identify Koopman eigenfunctions**. The overarching goal of this work is to leverage the power of deep learning to discover and represent eigenfunctions of the Koopman operator. Our perspective is driven by the need for parsimonious representations that are efficient, avoid overfitting, and provide minimal descriptions of the dynamics on interpretable intrinsic coordinates. Unlike previous deep learning approaches to Koopman[28–31], our network architecture is designed specifically

to handle a ubiquitous class of nonlinear systems characterized by a continuous frequency spectrum generated by the nonlinearity. A continuous spectrum presents unique challenges for compact and interpretable representation, and our approach is inspired by the classical asymptotic and perturbation approaches in dynamical systems.

Our core network architecture is shown in Fig. 1, and it is modified in Fig. 2 to handle the continuous spectrum. The objective of this network is to identify a few key intrinsic coordinates $\mathbf{y} = \varphi(\mathbf{x})$ spanned by a set of Koopman eigenfunctions $\varphi : \mathbb{R}^n \to \mathbb{R}^p$, along with a dynamical system $\mathbf{y}_{k+1} = \mathbf{K}\mathbf{y}_k$. There are three high-level requirements for the network, corresponding to three types of loss functions used in training:

1. Intrinsic coordinates that are useful for reconstruction. We seek to identify a few intrinsic coordinates $\mathbf{y} = \varphi(\mathbf{x})$ where the dynamics evolve, along with an inverse $\mathbf{x} = \varphi^{-1}(\mathbf{y})$ so that the state $\mathbf{x}$ may be recovered. This is achieved using an auto-encoder (see Fig. 1a), where $\varphi$ is the encoder and $\varphi^{-1}$ is the decoder. The dimension $p$ of the auto-encoder subspace is a hyperparameter of the network, and this choice may be guided by knowledge of the system. Reconstruction accuracy of the auto-encoder is achieved using the following loss: $\|\mathbf{x} - \varphi^{-1}(\varphi(\mathbf{x}))\|$.
2. Linear dynamics. To discover Koopman eigenfunctions, we learn the linear dynamics $\mathbf{K}$ on the intrinsic coordinates, i.e., $\mathbf{y}_{k+1} = \mathbf{K}\mathbf{y}_k$. Linear dynamics are achieved using the following loss: $\|\varphi(\mathbf{x}_{k+1}) - \mathbf{K}\varphi(\mathbf{x}_k)\|$. More generally, we enforce linear prediction over $m$ time steps with the loss: $\|\varphi(\mathbf{x}_{k+m}) - K^m\varphi(\mathbf{x}_k)\|$. (see Fig. 1c).
3. Future state prediction. Finally, the intrinsic coordinates must enable future state prediction. Specifically, we identify linear dynamics in the matrix $\mathbf{K}$. This corresponds to the loss $\|\mathbf{x}_{k+1} - \varphi^{-1}(\mathbf{K}\varphi(\mathbf{x}_k))\|$, and more generally $\|\mathbf{x}_{k+m} - \varphi^{-1}(\mathbf{K}^m\varphi(\mathbf{x}_k))\|$. (see Fig. 1b).

Our norm $\|\cdot\|$ is mean-squared error, averaging over dimension then number of examples, and we add $\ell_2$ regularization.

To address the continuous spectrum, we allow the eigenvalues of the matrix $\mathbf{K}$ to vary, parametrized by the function $\lambda = \Lambda(\mathbf{y})$, which is learned by an auxiliary network (see Fig. 2). The eigenvalues $\lambda_\pm = \mu \pm i\omega$ are then used to parametrize block-diagonal $\mathbf{K}(\mu,\omega)$. For each pair of complex eigenvalues, the discrete-time $\mathbf{K}$ has a Jordan block of the form:

$$B(\mu, \omega) = \exp(\mu\Delta t)\begin{bmatrix} \cos(\omega\Delta t) & -\sin(\omega\Delta t) \\ \sin(\omega\Delta t) & \cos(\omega\Delta t) \end{bmatrix}. \qquad (4)$$

This network structure allows the eigenvalues to vary across phase space, facilitating a small number of eigenfunctions. To enforce circular symmetry in the eigenfunction coordinates, we often parameterize the eigenvalues by the radius $\lambda(\|\mathbf{y}\|_2^2)$. The second and third prediction loss function must also be modified for systems with continuous spectrum, as discussed in the Methods section.

To train our network, we generate trajectories from random initial conditions, which are split into training, validation, and test sets. Models are trained on the training set and compared on the validation set, which is also used for early stopping to prevent overfitting. We report accuracy on the test set.

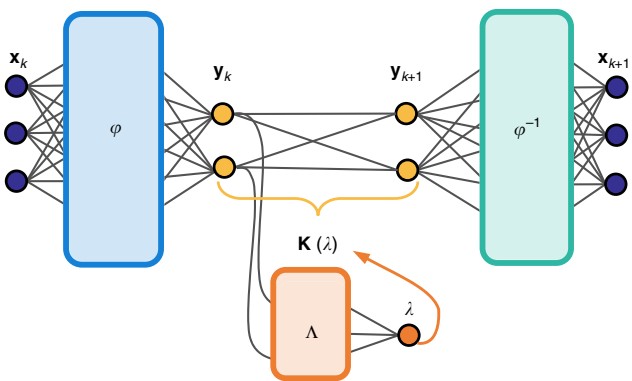

**Fig. 2** Schematic of modified schema with auxiliary network to identify (parametrize) the continuous eigenvalue spectrum $\lambda$. This facilitates an aggressive dimensionality reduction in the auto-encoder, avoiding the need for higher harmonics of the fundamental frequency that are generated by the nonlinearity[43, 44]. For purely oscillatory motion, as in the pendulum, we identify the continuous frequency $\lambda_\pm = \pm i\omega$

**Demonstration on examples**. We demonstrate our deep learning approach to identify Koopman eigenfunctions on several example systems, including a simple model with a discrete spectrum and two examples that exhibit a continuous spectrum: the nonlinear

| | Discrete spectrum | Pendulum | Fluid flow 1 | Fluid flow 2 |
|---|---|---|---|---|
| Training | $1.4 \times 10^{-7}$ | $8.5 \times 10^{-8}$ | $5.4 \times 10^{-7}$ | $2.8 \times 10^{-6}$ |
| Validation | $1.4 \times 10^{-7}$ | $9.4 \times 10^{-8}$ | $5.4 \times 10^{-7}$ | $2.9 \times 10^{-6}$ |
| Test | $1.5 \times 10^{-7}$ | $1.1 \times 10^{-7}$ | $5.5 \times 10^{-7}$ | $2.9 \times 10^{-6}$ |

**Table 1 Errors for each problem**

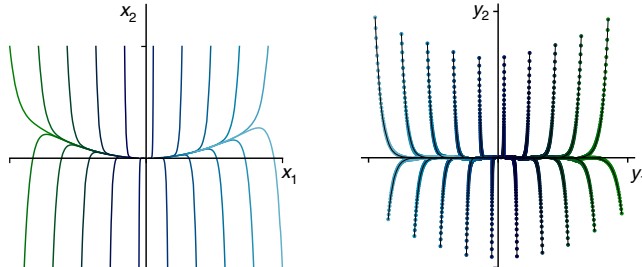

**Fig. 3** Demonstration of neural network embedding of Koopman eigenfunctions for simple system with a discrete eigenvalue spectrum

pendulum and the high-dimensional unsteady fluid flow past a cylinder. The training, validation, and test errors for all examples are reported in Table 1. Additional details for each example are provided in Supplementary Note 1.

**Example 1: Simple model with discrete spectrum.** Before analyzing systems with the additional challenges of a continuous spectrum and high-dimensionality, we consider a simple nonlinear system with a single fixed point and a discrete eigenvalue spectrum:

$$\dot{x}_1 = \mu x_1 \tag{5}$$

$$\dot{x}_2 = \lambda(x_2 - x_1^2). \tag{6}$$

This dynamical system has been well-studied in the literature[40,48], and for stable eigenvalues $\lambda < \mu < 0$, the system exhibits a slow manifold given by $x_2 = x_1^2$; we use $\mu = -0.05$ and $\lambda = -1$. As shown in Fig. 3, the Koopman embedding identifies nonlinear coordinates that flatten this inertial manifold, providing a globally linear representation of the dynamics; moreover, the correct Koopman eigenvalues are identified. Specific details about the network and training procedure are provided in the Methods.

In this example, we include the auxiliary network even though it is not required for examples with discrete eigenvalues. As shown in Supplementary Fig. 3, although the eigenvalues have the freedom to vary, they stay in a narrow range around the correct values. This numerically demonstrates that it is possible to identify a discrete spectrum without a priori knowledge about whether the spectrum is continuous or discrete.

**Example 2: Nonlinear pendulum with continuous spectrum.** As a second example, we consider the nonlinear pendulum, which exhibits a continuous eigenvalue spectrum with increasing energy:

$$\ddot{x} = -\sin(x) \quad \Rightarrow \quad \begin{cases} \dot{x}_1 = x_2 \\ \dot{x}_2 = -\sin(x_1). \end{cases} \tag{7}$$

Although this is a simple mechanical system, it has eluded

parsimonious representation in the Koopman framework. The deep Koopman embedding is shown in Fig. 4, where it is clear that the dynamics are linear in the eigenfunction coordinates, given by $\mathbf{y} = \varphi(\mathbf{x})$. As the Hamiltonian energy of the system increases, corresponding to an elongation of the oscillation period, the parameterized Koopman network accounts for this continuous frequency shift and provides a compact representation in terms of two conjugate eigenfunctions. Alternative network architectures that are not specifically designed to account for continuous spectra with an auxiliary network would be forced to approximate this frequency shift with the classical asymptotic expansion in terms of harmonics. The resulting network would be overly bulky and would limit interpretability.

Recall that we have three types of losses on the network: reconstruction, prediction, and linearity. Figure 4b shows that the network is able to function as an auto-encoder, accurately reconstructing the 10 example trajectories. Next, we show that the network is able to predict the evolution of the system. Figure 4c shows the prediction horizon for 10 initial conditions that are simulated forward with the network, stopping the prediction when the relative error reaches 10%. As expected, the prediction horizon deteriorates as the energy of the initial condition increases, although the prediction is still quite accurate. The nearly concentric circles in Fig. 4d demonstrate that the dynamics in the intrinsic coordinates $\mathbf{y}$ are truly linear.

In this example, both the eigenfunctions and the eigenvalues are spatially varying. When originally designing the Koopman network, we did not impose any constraints on how these eigenfunctions and eigenvalues vary in space, and the resulting network did not converge to a unique and interpretable solution. This led us to decide on an important design constraint, that a nonlinear oscillator, like the pendulum, should map to coordinates that have radial symmetry, so that the spatial variation of the eigenfunctions and eigenvalues depends on the radius of the intrinsic coordinates.

The eigenfunctions $\varphi_1(\mathbf{x})$ and $\varphi_2(\mathbf{x})$ are shown in Fig. 4e. It is possible to map these eigenfunctions into magnitude and phase coordinates, as shown in Fig. 5, where it can be seen that that magnitude essentially traces level sets of the Hamiltonian energy. This is consistent with previous theoretical derivations of Mezić[49] that represent Koopman eigenfunctions in action–angle coordinates, and we thank him for communicating this connection to us.

**Example 3: High-dimensional nonlinear fluid flow.** As our final example, we consider the nonlinear fluid flow past a circular cylinder at Reynolds number 100 based on diameter, which is characterized by vortex shedding. This model has been a benchmark in fluid dynamics for decades[50], and has been extensively analyzed in the context of data-driven modeling[25,51] and Koopman analysis[52]. In 2003, Noack et al.[50] showed that the high-dimensional dynamics evolve on a low-dimensional attractor, given by a slow-manifold in the following model:

$$\dot{x}_1 = \mu x_1 - \omega x_2 + A x_1 x_3 \tag{8}$$

$$\dot{x}_2 = \omega x_1 + \mu x_2 + A x_2 x_3 \tag{9}$$

$$\dot{x}_3 = -\lambda(x_3 - x_1^2 - x_2^2). \tag{10}$$

This mean-field model exhibits a stable limit cycle corresponding to von Karman vortex shedding, and an unstable equilibrium corresponding to a low-drag condition. Starting near this

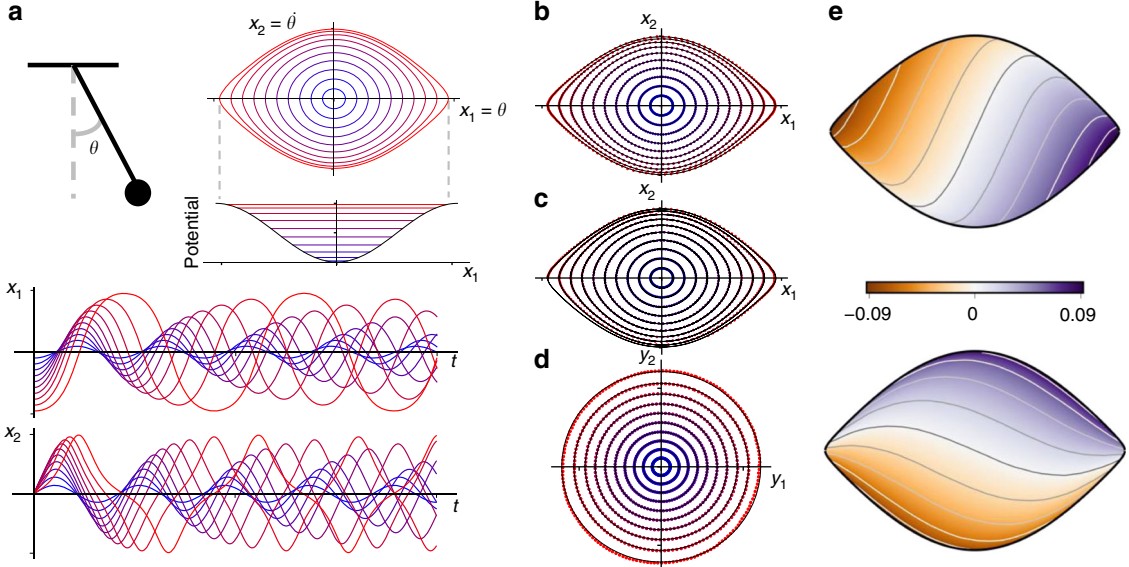

**Fig. 4** Illustration of deep Koopman eigenfunctions for the nonlinear pendulum. The pendulum, although a simple mechanical system, exhibits a continuous spectrum, making it difficult to obtain a compact representation in terms of Koopman eigenfunctions. However, by leveraging a generalized network, as in Fig. 2, it is possible to identify a parsimonious model in terms of a single complex conjugate pair of eigenfunctions, parameterized by the frequency $\omega$. In eigenfunction coordinates, the dynamics become linear, and orbits are given by perfect circles. For the sake of visualization, we use 10 evenly spaced trajectories instead of the random trajectories in the testing set

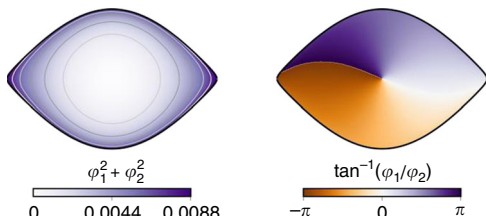

**Fig. 5** Magnitude and phase of the pendulum eigenfunctions

equilibrium, the flow unwinds up the slow manifold toward the limit cycle. In[51], Loiseau and Brunton showed that this flow may be modeled by a nonlinear oscillator with state-dependent damping, making it amenable to the continuous spectrum analysis. We use trajectories from this model where $\mu = 0.1$, $\omega = 1$, $A = -0.1$, and $\lambda = 10$ to train a Koopman network. The resulting eigenfunctions are shown in Fig. 6.

In this example, the damping rate $\mu$ and frequency $\omega$ are allowed to vary along level sets of the radius in eigenfunction coordinates, so that $\mu(R)$ and $\omega(R)$, where $R^2 = y_1^2 + y_2^2$; this is accomplished with an auxiliary network as in Fig. 2. These continuously varying eigenvalues are shown in Supplementary Fig. 5, where it can be seen that the frequency $\omega$ is extremely close to the true constant $-1$, while the damping $\mu$ varies significantly, and in fact switches stability for trajectories outside the natural limit cycle. This is consistent with the data-driven model of Loiseau and Brunton[51].

Although we only show the ability of the model to predict the future state in Fig. 6, corresponding to the second and third loss functions, the network also functions as an autoencoder. Figure 6c shows the prediction performance of the Koopman network for trajectories that start away from the attractor; in both cases, the dynamics are faithfully captured and the dynamics attract onto the limit cycle. Thus, it is possible to

capture nonlinear transients, as long as these are sufficiently represented in the training data.

## Discussion

In summary, we have employed powerful deep learning approaches to identify and represent coordinate transformations that recast strongly nonlinear dynamics into a globally linear framework. Our approach is designed to discover eigenfunctions of the Koopman operator, which provide an intrinsic coordinate system to linearize nonlinear systems, and have been notoriously difficult to identify and represent using alternative methods. Building on a deep auto-encoder framework, we enforce additional constraints and loss functions to identify Koopman eigenfunctions where the dynamics evolve linearly. Moreover, we generalize this framework to include a broad class of nonlinear systems that exhibit a continuous eigenvalue spectrum, where a continuous range of frequencies is observed. Continuous-spectrum systems are notoriously difficult to analyze, especially with Koopman theory, and naive learning approaches require asymptotic expansions in terms of higher order harmonics of the fundamental frequency, leading to unwieldy models. In contrast, we utilize an auxiliary network to parametrize and identify the continuous frequency, which then parameterizes a compact Koopman model on the auto-encoder coordinates. Thus, our deep neural network models remain both parsimonious and interpretable, merging the best of neural network representations and Koopman embeddings. In most deep learning applications, although the basic architecture is extremely general, considerable expert knowledge and intuition is typically used in the training process and in designing loss functions and constraints. Throughout this paper, we have also used physical insight and intuition from asymptotic theory and continuous spectrum dynamical systems to guide the design of parsimonious Koopman embeddings.

There are many ongoing challenges and promising directions that motivate future work. First, there are still several limitations associated with deep learning, including the need for vast and diverse data and extensive computation to train models[53]. This

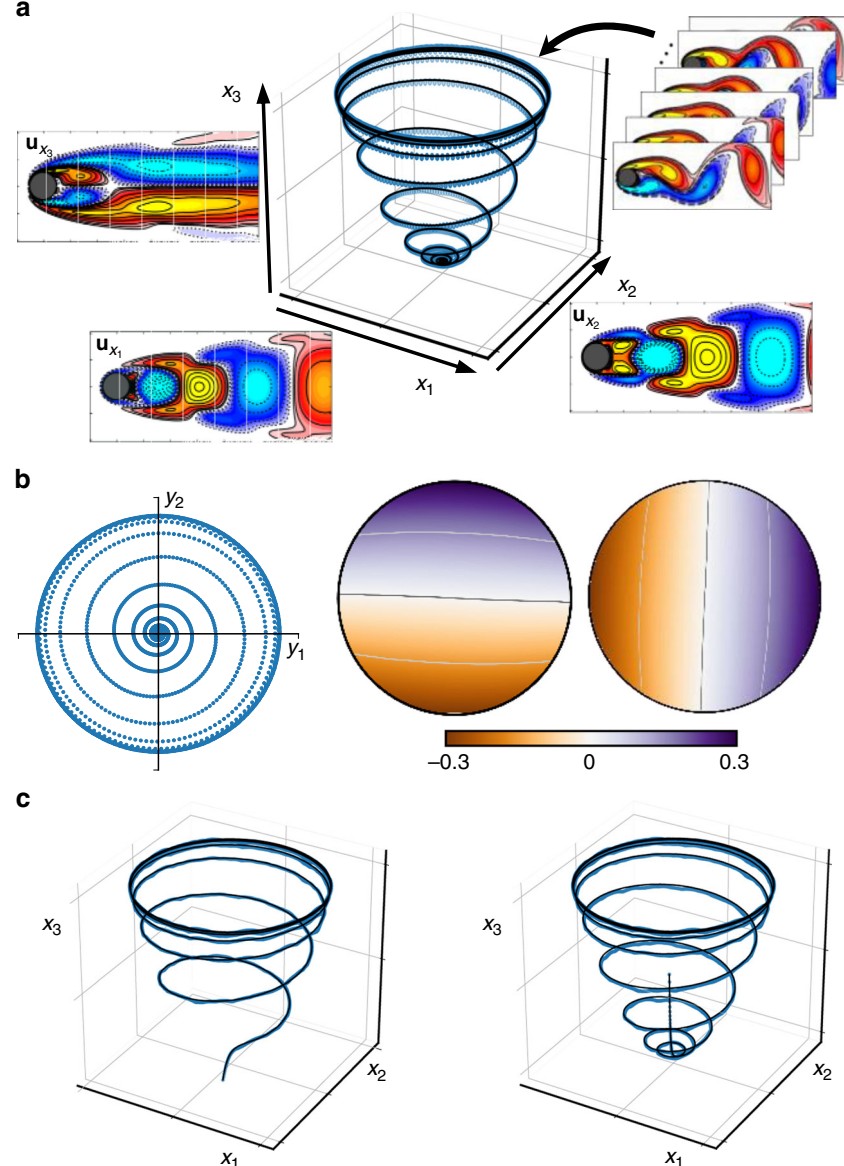

**Fig. 6** Learned Koopman eigenfunctions for the mean-field model of fluid flow past a circular cylinder at Reynolds number 100. **a** Reconstruction of trajectory from linear Koopman model with two states; modes for each of the state space variables $x$ are shown along the coordinate axes. **b** Koopman reconstruction in eigenfunction coordinates $y$, along with eigenfunctions $y = \varphi(x)$. **c** Two examples of trajectories that begin off the attractor. The Koopman model is able to reconstruct both given only the initial condition

training may be considered a one-time upfront cost, and deep learning frameworks, such as TensorFlow parallelize the training on GPUs and across GPUs[54]; further, there is ongoing work to improve the scalability[53]. Even more concerning is the dubious generalizability and interpretability of the resulting models, as deep learning architectures may be viewed as sophisticated interpolation engines with limited ability to extrapolate beyond the training data[55]. This work attempts to promote interpretability by forcing the network to have physical meaning in the context of Koopman theory, although the issue with generalizability still requires sufficient volumes and diversity of training data. There are also more specific limitations to the current proposed architecture, foremost, choosing the dimension of the autoencoder coordinates, **y**. Continued effort will be required to automatically detect the dimension of the intrinsic coordinates and to classify spectra (e.g., discrete and continuous, and real and

complex eigenvalues). It will be important to extend these methods to higher-dimensional examples with more complex energy spectra, as the examples considered here are relatively low-dimensional. Fortunately, with sufficient data, deep learning architectures are able to learn incredibly complex representations, so the prospects for scaling these methods to larger systems is promising.

The use of deep learning in physics and engineering is increasing at an incredible rate, and this trend is only expected to accelerate. Nearly every field of science is revisiting challenging problems of central importance from the perspective of big data and deep learning. With this explosion of interest, it is imperative that we as a community seek machine learning models that favor interpretability and promote physical insight and intuition. In this challenge, there is a tremendous opportunity to gain new understanding and insight by applying increasingly powerful techniques to data. For example,

discovering Koopman eigenfunctions will result in new symmetries and conservation laws, as conserved eigenfunctions are related to conservation laws via a generalized Noether's theorem. It will also be important to apply these techniques to increasingly challenging problems, such as turbulence, epidemiology, and neuroscience, where data is abundant and models are needed. The goal is to model these systems with a small number of coupled nonlinear oscillators using similar parameterized Koopman embeddings. Finally, the use of deep learning to discover Koopman eigenfunctions may enable transformative advances in the nonlinear control of complex systems. All of these future directions will be facilitated by more powerful network representations.

## Methods

**Creating the datasets**. We create our datasets by solving the systems of differential equations in MATLAB using the ode45 solver.

For each dynamical system, we choose 5000 initial conditions for the test set, 5000 for the validation set, and 5000–20,000 for the training set (see Table 2). For each initial condition, we solve the differential equations for some time span. That time span is $t = 0, .02, \ldots, 1$ for the discrete spectrum and pendulum datasets. Since the dynamics on the slow manifold for the fluid flow example are slower and more complicated, we increase the time span for that dataset to $t = 0, .05, \ldots, 6$. However, when we include data off the slow manifold, we want to capture the fast dynamics as the trajectories are attracted to the slow manifold, so we change the time span to $t = 0, .01, \ldots, 1$. Note that for the network to capture transient behavior as in the first and last example, it is important to include enough samples of transients in the training data.

The discrete spectrum dataset is created from random initial conditions $\mathbf{x}$ where $x_1, x_2 \in [-0.5, 0.5]$, since this portion of phase space is sufficient to capture the dynamics.

The pendulum dataset is created from random initial conditions $\mathbf{x}$, where $x_1 \in [-3.1, 3.1]$ (just under $[-\pi, \pi]$), $x_2 \in [-2, 2]$, and the potential function is under 0.99. The potential function for the pendulum is $\frac{1}{2}x_2^2 - \cos(x_1)$. These ranges are chosen to sample the pendulum in the full phase space where the pendulum approaches having an infinite period.

The fluid flow problem limited to the slow manifold is created from random initial conditions $\mathbf{x}$ on the bowl where $r \in [0, 1.1]$, $\theta \in [0, 2\pi]$, $x_1 = r\cos(\theta)$, $x_2 = r\sin(\theta)$, and $x_3 = x_1^2 + x_2^2$. This captures all of the dynamics on the slow manifold, which consists of trajectories that spiral toward the limit cycle at $r = 1$.

The fluid flow problem beyond the slow manifold is created from random initial conditions $\mathbf{x}$ where $x_1 \in [-1.1, 1.1]$, $x_2 \in [-1.1, 1.1]$, and $x_3 \in [0, 2.42]$. These limits are chosen to include the dynamics on the slow manifold covered by the previous dataset, as well as trajectories that begin off the slow manifold. Any trajectory that grows to $x_3 > 2.5$ is eliminated so that the domain is reasonably compact and well-sampled.

**Code**. We use the Python API for the TensorFlow framework[54] and the Adam optimizer[56] for training. All of our code is available online at github.com/BethanyL/DeepKoopman.

**Network architecture**. Each hidden layer has the form of $\mathbf{Wx} + \mathbf{b}$ followed by an activation with the rectified linear unit (ReLU): $f(\mathbf{x}) = \max\{0, \mathbf{x}\}$. In our

### Table 2 Dataset sizes

|  | Discrete spectrum | Pendulum | Fluid flow 1 | Fluid flow 2 |
|---|---|---|---|---|
| Length of traj. | 51 | 51 | 121 | 101 |
| # Training traj. | 5000 | 15,000 | 15,000 | 20,000 |
| Batch size | 256 | 128 | 256 | 128 |

### Table 3 Network architecture

|  | Discrete spectrum | Pendulum | Fluid flow 1 | Fluid flow 2 |
|---|---|---|---|---|
| # hidden layers (HL) | 2 | 2 | 1 | 1 |
| Width HL | 30 | 80 | 105 | 130 |
| # HL aux. net. | 3 | 1 | 1 | 2 |
| Width HL aux. net. | 10 | 170 | 300 | 20 |

### Table 4 Loss hyperparameters

|  | Discrete spectrum | Pendulum | Fluid flow 1 | Fluid flow 2 |
|---|---|---|---|---|
| $\alpha_1$ | 0.1 | 0.001 | 0.1 | 0.1 |
| $\alpha_2$ | $10^{-7}$ | $10^{-9}$ | $10^{-7}$ | $10^{-9}$ |
| $\alpha_3$ | $10^{-15}$ | $10^{-14}$ | $10^{-13}$ | $10^{-13}$ |
| $S_p$ | 30 | 30 | 30 | 30 |

experiments, training was significantly faster with ReLU as the activation function than with sigmoid. See Table 3 for the number of hidden layers in the encoder, decoder, and auxiliary network, as well as their widths. The output layers of the encoder, decoder, and auxiliary network are linear (simply $\mathbf{Wx} + \mathbf{b}$).

The input to the auxiliary network is $y$, and it outputs the parameters for the eigenvalues of $\mathbf{K}$. For each complex conjugate pair of eigenvalues $\lambda_\pm = \mu \pm i\omega$, the network defines a function $\Lambda$ mapping $y_j^2 + y_{j+1}^2$ to $\mu$ and $\omega$, where $y_j$ and $y_{j+1}$ are the corresponding eigenfunctions. Similarly, for each real eigenvalue $\lambda$, the network defines a function mapping $y_j$ to $\lambda$. For example, for the fluid flow problem off the attractor, we have three eigenfunctions. The auxiliary network learns a map from $y_1^2 + y_2^2$ to $\mu$ and $\omega$ and another map from $y_3$ to $\lambda$. This could be implemented as one network defining a mapping $\Lambda : \mathbb{R}^2 \to \mathbb{R}^3$ where the layers are not fully connected ($y_1^2 + y_2^2$ should not influence $y_3$ and $y_3$ should not influence $\mu$ and $\omega$). However, for simplicity, we implement this as two separate auxiliary networks, one for the complex conjugate pair of eigenvalues and one for the the real eigenvalue.

**Explicit loss function**. Our loss function has three weighted mean-squared error components: reconstruction accuracy $\mathcal{L}_{\text{recon}}$, future state prediction $\mathcal{L}_{\text{pred}}$, and linearity of dynamics $\mathcal{L}_{\text{lin}}$. Since we know that there are no outliers in our data, we also use an $\mathcal{L}_\infty$ term to penalize the data point with the largest loss. Finally, we add $\ell_2$ regularization on the weights $W$ to avoid overfitting. More specifically:

$$\mathcal{L} = \alpha_1 (\mathcal{L}_{\text{recon}} + \mathcal{L}_{\text{pred}}) + \mathcal{L}_{\text{lin}} + \alpha_2 \mathcal{L}_\infty + \alpha_3 \|\mathbf{W}\|_2^2 \quad (11)$$

$$\mathcal{L}_{\text{recon}} = \left\| \mathbf{x}_1 - \varphi^{-1}(\varphi(\mathbf{x}_1)) \right\|_{\text{MSE}} \quad (12)$$

$$\mathcal{L}_{\text{pred}} = \frac{1}{S_p} \sum_{m=1}^{S_p} \left\| \mathbf{x}_{m+1} - \varphi^{-1}(K^m \varphi(\mathbf{x}_1)) \right\|_{\text{MSE}} \quad (13)$$

$$\mathcal{L}_{\text{lin}} = \frac{1}{T-1} \sum_{m=1}^{T-1} \left\| \varphi(\mathbf{x}_{m+1}) - K^m \varphi(\mathbf{x}_1) \right\|_{\text{MSE}} \quad (14)$$

$$\mathcal{L}_\infty = \left\| \mathbf{x}_1 - \varphi^{-1}(\varphi(\mathbf{x}_1)) \right\|_\infty + \left\| \mathbf{x}_2 - \varphi^{-1}(K\varphi(\mathbf{x}_1)) \right\|_\infty, \quad (15)$$

where MSE refers to mean squared error and $T$ is the number of time steps in each trajectory. The weights $\alpha_1$, $\alpha_2$, and $\alpha_3$ are hyperparameters. The integer $S_p$ is a hyperparameter for how many steps to check in the prediction loss. The hyperparameters $\alpha_1$, $\alpha_2$, $\alpha_3$, and $S_p$ are listed in Table 4.

The matrix $\mathbf{K}$ is parametrized by the function $\lambda = \Lambda(\mathbf{y})$, which is learned by an auxiliary network. The eigenvalues can vary along a trajectory, so in $\mathcal{L}_{\text{pred}}$ and $\mathcal{L}_{\text{lin}}$, $\mathbf{K}^m = \mathbf{K}(\lambda_1) \cdot \mathbf{K}(\lambda_2) \ldots \mathbf{K}(\lambda_m)$. However, on Hamiltonian systems, such as the pendulum, the eigenvalues are constant along each trajectory. If a system is known to be Hamiltonian, the network training could be sped up by encoding the constraint that $\mathbf{K}^m = \mathbf{K}(\lambda)^m$. In order to demonstrate that this specialized knowledge is not necessary, we use the more general case for all of our datasets, including the pendulum.

**Training**. We initialize each weight matrix $W$ randomly from a uniform distribution in the range $[-s, s]$ for $s = 1/\sqrt{a}$, where $a$ is the dimension of the input of the layer. This distribution was suggested in ref. [21]. Each bias vector $\mathbf{b}$ is initialized to 0. The model for the discrete spectrum example is trained for 2 h on an NVIDIA K80 GPU. The other models are each trained for 6 h. The learning rate for the Adam optimizer is 0.001. On the pendulum and fluid flow datasets, for 5 min, we pretrain the network to be a simple autoencoder, using the autoencoder loss but not the linearity or prediction losses, as this speeds up the training. We also use early stopping; for each model, at the end of training, we resume the step with the lowest validation error.

**Hyperparameter tuning**. There are many design choices in deep learning, so we use hyperparameter tuning, as described in ref. [21]. For each dynamical system, we train multiple models in a random search of hyperparameter space and choose the one with the lowest validation error. Each model is also initialized with different

random weights. We find that $\alpha_1$, which defines a trade-off between the two objectives that include the decoder and the one that does not, has a significant effect on the training speed.

**Code availability**. All code used in this study is available at github.com/BethanyL/DeepKoopman.

## Data availability
All data generated during this study can be reconstructed using the code available at github.com/BethanyL/DeepKoopman.

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

## Acknowledgements

We acknowledge generous funding from the Army Research Office (ARO W911NF-17-1-0306) and the Defense Advanced Research Projects Agency (DARPA HR0011-16-C-0016). We would like to thank many people for valuable discussions about neural networks and Koopman theory: Bing Brunton, Karthik Duraisamy, Eurika Kaiser, Bernd Noack, and Josh Proctor, and especially Jean-Christophe Loiseau, Igor Mezić, and Frank Noé.

## Author contributions

B.L. performed research; B.L., J.N.K., and S.L.B. designed research, analyzed data, and wrote the paper.

## Additional information

**Competing interests:** The authors declare no competing interests.

