## [Peer Review File · Nature Communications]

Reviewers' comments:

Reviewer #1 (Remarks to the Author):

The idea of the auxiliary neural network is really appealing. Very interesting and novel work! The authors are also experts in the field with a long list of publications. I recommend publications.

My only suggestions are:

1) Although the results presented are important, I think it is fair to say that a 3-dimensional model of the flow behind the cylinder for $Re=100$ is not an accurate representative of "broadband turbulence". I think the statement in the abstract and elsewhere need to be more accurate.

2) It would be important to emphasize the limitations of the method, at least from the point of view of what systems can/is expected this approach to handle, i.e. in terms of dimensionality and in terms of energy spectrum since all the examples considered have relatively low dimensionality.

Reviewer #2 (Remarks to the Author):

In this paper, the authors propose a deep NN model for identifying a coordinate that globally linearizes nonlinear dynamics. And, they demonstrate the model with simulation data in several scenarios. I enjoyed reading this paper, which discusses a relevant application of deep learning to the operator-theoretic analysis of nonlinear dynamical systems. I think that the paper provides useful insights in the topic and would be of interest for many readers.

On the other hand, I have some technical concerns, especially, about the part for handling continuous spectra, which seems one of the key features of their model. In the model, the authors incorporate an auxiliary network that directly parameterizes continuous spectra, which affects the transition matrix from y_k to y_{k+1} . This part could help to overcome the weakness of DMD that in principle assumes distinct spectra. However, from the learning perspective, it seems not easy to correctly distinguish the behaviors of distinct and continuous spectra only from data in their model. In particular, this looks affected via a complicated hyper-parameter tuning process (Eq.(4) in the supplementary document), which would not be easy to give even a guideline for the tuning. In fact, it seems difficult to see some tendency in Table 3.

Also, I wonder how the relation between Koopman eigenfunctions and (varying) eigenvalues are held (it seems likely that the eigenfunctions also should change when the corresponding eigenvalues change?).

And, although it is related to the above, I think that the paper does not clearly describe possible limitations of their model or the assumptions under their model. For example, what kind of transient dynamics could be solved with the proposed model. Or, how is the computational cost for learning the model?

As a summary, although this paper discuss a relevant application of deep learning to data-driven analysis of dynamical systems, it has some technically unclear points that should be mentioned more explicitly. I am not necessarily positive for publication of this paper at this journal but I have no objection to that. However, I think modifications of the paper addressing the above concerns would be mandatory.

Response to Reviewers

for "Deep learning for universal linear embeddings of nonlinear dynamics" by B. Lusch, J. N. Kutz, and S. L. Brunton

We thank the reviewers for their detailed and constructive review of our manuscript. We have made every attempt to address these comments, and we believe that this has strengthened our manuscript. In what follows, we provide responses to all concerns raised by the reviewers (in blue).

Specific additions to the manuscript are shown in red.

In the paper, changes are red and text that was moved from the supplement into the main paper is blue.

Reviewer #1:

The idea of the auxiliary neural network is really appealing. Very interesting and novel work! The authors are also experts in the field with a long list of publications. I recommend publications.

Thank you for your positive assessment of our work! The comments below are also very helpful.

My only suggestions are:

1) Although the results presented are important, I think it is fair to say that a 3-dimensional model of the flow behind the cylinder for $Re=100$ is not an accurate representative of "broadband turbulence". I think the statement in the abstract and elsewhere need to be more accurate.

We completely agree and have removed this wording to be more precise.

2) It would be important to emphasize the limitations of the method, at least from the point of view of what systems can/is expected this approach to handle, i.e. in terms of dimensionality and in terms of energy spectrum since all the examples considered have relatively low dimensionality.

This is a very good point, and we have expanded the discussion to discuss and emphasize the limitations of the method. In particular, we focus on current limitations of deep learning, in general, and also on limitations of the proposed approach to discover and represent Koopman eigenfunctions. This discussion has been added to the manuscript:

There are many ongoing challenges and promising directions that motivate future work. First, there are still several limitations associated with deep learning, including the need for vast and diverse data and extensive computation to train models [3]. This training may be considered a one-time upfront cost, and deep learning frameworks such as TensorFlow parallelize the training on GPUs and across GPUs [2]; further, there is ongoing work to improve the scalability [3]. Even more concerning is the dubious generalizability and interpretability of the resulting models, as deep learning architectures may be viewed as sophisticated interpolation engines with limited ability to extrapolate beyond the training data [5]. This work attempts to promote interpretability by forcing the network to have physical meaning in the context of Koopman theory, although the issue with generalizability still requires sufficient volumes and diversity of training data. There are also more specific limitations to the current proposed architecture, foremost, choosing the dimension of the autoencoder coordinates, y . Continued effort will be required to automatically detect the dimension of the intrinsic coordinates and to classify spectra (e.g., discrete and continuous, and real and complex eigenvalues). It will be important to extend these methods to higher-dimensional examples with more complex energy spectra, as the examples considered here are relatively low-dimensional. Fortunately, with sufficient data, deep learning architectures are able to learn incredibly complex representations, so the prospects for scaling these methods to larger systems is promising.

Reviewer #2:

In this paper, the authors propose a deep NN model for identifying a coordinate that globally linearizes nonlinear dynamics. And, they demonstrate the model with simulation data in several scenarios. **I enjoyed reading this paper, which discusses a relevant application of deep learning to the operator-theoretic analysis of nonlinear dynamical systems. I think that the paper provides useful insights in the topic and would be of interest for many readers.**

We would like to thank the referee for recognizing that this paper is useful and interesting for readers. We also greatly appreciate the constructive and insightful comments below.

On the other hand, I have some technical concerns, especially, about the part for handling continuous spectra, which seems one of the key features of their model. In the model, the authors incorporate an auxiliary network that directly parameterizes continuous spectra, which affects the transition matrix from y_k to y_{k+1} . This part could help to overcome the weakness of DMD that in principle assumes distinct spectra. However, from the learning perspective, it seems not easy to correctly distinguish the behaviors of distinct and continuous spectra only from data in their model.

This is an excellent point, and something that we also had initial concerns about. We were concerned that the user would need to know *a priori* if the system had a continuous or discrete spectrum. The auxiliary network is only necessary for examples with continuous spectra, but we also included this auxiliary network on the first example with a discrete spectrum. Even though the eigenvalues had the freedom to vary, they stayed in a very small range, which numerically indicates a discrete spectrum. (See Supplementary Figure 3.) This demonstrates that it is not necessary to know ahead of time whether or not the data has continuous spectra. We have clarified this important point in the manuscript, as this is an important validation step. Thank you for bringing this up.

In this example, we include the auxiliary network even though it is not required for examples with discrete eigenvalues. As shown in Supplementary Figure 3, although the eigenvalues have the freedom to vary, they stay in a narrow range around the correct values. This numerically demonstrates that it is possible to identify a discrete spectrum without *a priori* knowledge about whether the spectrum is continuous or discrete.

In particular, this looks affected via a complicated hyper-parameter tuning process (Eq.(4) in the supplementary document), which would not be easy to give even a guideline for the tuning. In fact, it seems difficult to see some tendency in Table 3.

There are many design choices in deep learning, and you are absolutely correct that hyper-parameter tuning is necessary. We find that α_1 , which defines a trade-off between the two objectives that include the decoder and the one that does not, has a significant effect on the training speed. More generally, we follow the advice in [4] for conducting a random search of hyperparameter space. We use the validation error to compare trained networks. Our code for conducting a random hyperparameter search is included at github.com/BethanyL/DeepKoopman. We have moved much of these details into the methods of the main manuscript to make them easier to find. We have also included a new subsection to explicitly acknowledge the importance and challenge of hyper-parameter tuning.

However, an important point is that the auxiliary network may actually *simplify* network design and tuning, as we no longer rely on a harmonic expansion to approximate the continuous spectrum.

There are many design choices in deep learning, so we use hyperparameter tuning, as described in [4]. For each dynamical system, we train multiple models in a random search of hyperparameter space and choose the one with the lowest validation error. Each model is also initialized with different random weights. We find that α_1 , which defines a trade-off between the two objectives that include the decoder and the one that does not, has a significant effect on the training speed.

Also, I wonder how the relation between Koopman eigenfunctions and (varying) eigenvalues are held (it seems likely that the eigenfunctions also should change when the corresponding eigenvalues change?).

This is something that we wrestled with for a while in the design of our network. We originally designed a network with too many degrees of freedom (i.e., varying eigenvalues and eigenfunctions), and found that the network didn't have enough constraints to arrive at a unique and interpretable solution. We decided that one important design criteria (constraint) is that a nonlinear oscillator, like the pendulum, should map to coordinates that have a radial symmetry. We later learned, through a discussion with Igor Mezic, about a recent paper that explores how to interpret the Koopman operator for systems with continuous spectra in terms of action-angle coordinates. The eigenfunctions learned in our network are consistent with this action-angle perspective. We have also discussed this with Dimitris Giannakis, who has suggested that the continuous eigenvalue spectrum may be viewed as a rescaling of time. To our knowledge, the universality of this relationship is still the subject of ongoing research, and we hope that these theoretical aspects will continue to be developed by colleagues.

We have expanded our discussion to include these subtle points about the varying eigenvalues and fixed eigenfunctions:

In this example, both the eigenfunctions and the eigenvalues are spatially varying. When originally designing the Koopman network, we didn't impose any constraints on how these eigenfunctions and eigenvalues vary in space, and the resulting network didn't converge to a unique and interpretable solution. This led us to decide on an important design constraint, that a nonlinear oscillator, like the pendulum, should map to coordinates that have radial symmetry, so that the spatial variation of the eigenfunctions and eigenvalues depends on the radius of the intrinsic coordinates.

The eigenfunctions $\phi_1(x)$ and $\phi_2(x)$ are shown in Fig. 4(e). It is possible to map these eigenfunctions into magnitude and phase coordinates, as shown in Fig. 5, where it can be seen that that magnitude essentially traces level sets of the Hamiltonian energy. This is consistent with previous theoretical derivations of Mezic [1] that represent Koopman eigenfunctions in action-angle coordinates, and we thank him for communicating this connection to us.

And, although it is related to the above, I think that the paper does not clearly describe possible limitations of their model or the assumptions under their model.

This is an important point, and we have expanded the discussion to include several limitations of the method. These limitations are related to deep learning in general, as well as to this specific application of discovering Koopman eigenfunctions:

There are many ongoing challenges and promising directions that motivate future work. First, there are still several limitations associated with deep learning, including the need for vast and diverse data and extensive computation to train models [3]. This training may be considered a one-time upfront cost, and deep learning frameworks such as TensorFlow parallelize the training on GPUs and across GPUs [2]; further, there is ongoing work to improve the scalability [3]. Even more concerning is the dubious generalizability and interpretability of the resulting models, as deep learning architectures may be viewed as sophisticated interpolation engines with limited ability to extrapolate beyond the training data [5]. This work attempts to promote interpretability by forcing the network to have physical meaning in the context of Koopman theory, although the issue with generalizability still requires sufficient volumes and diversity of training data. There are also more specific limitations to the current proposed architecture, foremost, choosing the dimension of the autoencoder coordinates, y . Continued effort will be required to automatically detect the dimension of the intrinsic coordinates and to classify spectra (e.g., discrete and continuous, and real and complex eigenvalues). It will be important to extend these methods to higher-dimensional examples with more complex energy spectra, as the examples considered here are relatively low-dimensional. Fortunately, with sufficient data, deep learning architectures are able to learn incredibly complex representations, so the prospects for scaling these methods to larger systems is promising.

For example, what kind of transient dynamics could be solved with the proposed model.

We agree with the referee on the importance (and challenge) of capturing transient dynamics. We include two models with transient dynamics. Our first example (the discrete spectrum example) and our last example (the fluid flow) include trajectories that begin off the attractor. In both cases, we find that the training could learn the transient behavior if we included enough samples in the training data from the transients. (See, for example, Supplemental Figure 9.) As mentioned in the “Creating the datasets” section under Methods, we take this into consideration when choosing the time span for the sampling. However, it is also true that if transients are not included in the training data, it is unlikely that the network model will generalize to capture transients. We have addressed this important point more explicitly in the manuscript by adding more to the “Creating the datasets” subsection.

Note that for the network to capture transient behavior such as appears in the first and last example, it is important to include enough samples of the transients in the training data.

We also moved part of a figure from the supplement to the main paper that shows how the network handles prediction of transient behavior in the fluid flow problem. These prediction plots were previously the second row of Figure 9 in the supplement (now called Supplementary Figure 8) but are now row 3 of Figure 6 in the main paper. We also moved the corresponding description of the plots:

Fig. 6 (c) shows the prediction performance of the Koopman network for trajectories that start away from the attractor; in both cases, the dynamics are faithfully captured and the dynamics attract onto the

limit cycle. Thus, it is possible to capture nonlinear transients, as long as these are sufficiently represented in the training data.

We added further description of Supplementary Figure 8:

Supplementary Figure 8 shows the average prediction error versus the number of steps. Although the loss function only penalized 30 prediction steps in the future ($S_p = 30$), the error remains small for all 100 steps. The figure also shows the embedding of a trajectory in y coordinates. Although this network's training data includes data off the attractor, this network's embedding is similar to the embedding from the previous case. (See Figure 5 in the main paper.)

Figure 6: Learned Koopman eigenfunctions for the mean-field model of fluid flow past a circular cylinder at Reynolds number 100. (a) Reconstruction of trajectory from linear Koopman model with two states; modes for each of the state space variables x are shown along the coordinate axes. (b) Koopman reconstruction in eigenfunction coordinates y , along with eigenfunctions $y = \varphi(x)$. (c) Two examples of trajectories that begin off the attractor. The Koopman model is able to reconstruct both given only the initial condition.

Or, how is the computational cost for learning the model?

This is another important point. The training took 2-6 hours per example. We moved the information about the training time from the supplement to the Methods section of the main paper. We also added thoughts on the complexity of deep learning to the Discussion. (See the red text in response above.)

As a summary, although this paper discusses a relevant application of deep learning to data-driven analysis of dynamical systems, it has some technically unclear points that should be mentioned more explicitly. I am not necessarily positive for publication of this paper at this journal but I have no objection to that. However, I think modifications of the paper addressing the above concerns would be mandatory.

We would like to thank the referee again for making these excellent clarifying questions. We believe that this has helped us resolve many potentially unclear points and strengthen the manuscript. We hope that the referee agrees.

References

- [1] Mezić, I., "Koopman Operator Spectrum and Data Analysis," arXiv preprint arXiv:1702.07597, 2017.
- [2] Abadi, M. *Et al.*, "TensorFlow: Large-scale machine learning on heterogeneous systems," <http://tensorflow.org/>, 2015.
- [3] Cui, H., Zhang, H., Ganger, G. R., Gibbons, P. B. & Xing, E. P., "GeePS: Scalable deep learning on distributed GPUs with a GPU-specialized parameter server," In *Proceedings of the Eleventh European Conference on Computer Systems*, 4, 2016.
- [4] Goodfellow, I., Bengio, Y. & Courville, A. *Deep Learning*, MIT Press, 2016.
- [5] Mallat, S. "Understanding deep convolutional networks," *Phil. Trans. R. Soc. A* **374**, 20150203, 2016.

REVIEWERS' COMMENTS:

Reviewer #1 (Remarks to the Author):

The authors have addressed all my comments and to this end I recommend publication.

Reviewer #2 (Remarks to the Author):

Thank you for revising the paper carefully according to my comments. The revised parts make it convincing sufficiently for publication in this journal.

Response to Reviewers

for "Deep learning for universal linear embeddings of nonlinear dynamics" by B. Lusch, J. N. Kutz, and S. L. Brunton

Reviewer #1 (Remarks to the Author):

The authors have addressed all my comments and to this end I recommend publication.

Reviewer #2 (Remarks to the Author):

Thank you for revising the paper carefully according to my comments. The revised parts make it convincing sufficiently for publication in this journal.

Thank you for the positive feedback and for the chance to publish in *Nature Communications*.